# The importance of motivation in selecting undergraduate medical students for extracurricular research programmes

**Belinda W. C. Ommering**[1]*, **Floris M. Van Blankenstein**[1], **Merel van Diepen**[2], **Nelleke A. Gruis**[3], **Ada Kool**[4], **Friedo W. Dekker**[1,2]

**1** Center for Innovation in Medical Education, Leiden University Medical Center, Leiden, the Netherlands,
**2** Department of Clinical Epidemiology, Leiden University Medical Center, Leiden, the Netherlands,
**3** Department of Dermatology, Leiden University Medical Center, Leiden, the Netherlands, **4** Department of Educational Consultancy & Professional Development, Faculty of Social and Behavioral Sciences, Utrecht University, Utrecht, the Netherlands

* b.w.c.ommering@lumc.nl

## Abstract

### Introduction

Extracurricular research programmes (ERPs) may contribute to reducing the current shortage in physician-scientists, but usually select students based on grades only. The question arises if students should be selected based on their motivation, regardless of their previous academic performance. Focusing on grades and lacking to take motivation into account when selecting students for ERPs might exclude an important target group when aiming to cultivate future physician-scientists. Therefore, this study compared ERP students with lower and higher previous academic performance on subsequent academic performance, ERP performance, and motivational factors.

### Methods

Prospective cohort study with undergraduate medical students who filled in a yearly questionnaire on motivational factors. Two student groups participating in an ERP were compared: students with first-year grade point average (GPA) $\geq 7$ versus $<7$ on a 10-point grading scale. Linear and logistic regressions analyses were used to compare groups on subsequent academic performance (i.e. third-year GPA, in-time bachelor completion), ERP performance (i.e. drop-out, number of credits), and motivational factors (i.e. intrinsic motivation for research, research self-efficacy beliefs, perceptions of research, curiosity), while adjusting for gender and motivational factors at baseline.

### Results

The $<7$ group had significantly lower third-year GPA, and significantly higher odds for ERP drop-out than the $\geq 7$ group. However, there was no significant between-group difference on in-time bachelor completion and the $<7$ group was not inferior to the $\geq 7$ group in terms of intrinsic motivation for research, perceptions of research, and curiosity.

**Data Availability Statement:** Data cannot be shared publicly because in the consent form, we informed the participating students that data would

not be traceable to individual participants. As this is a somewhat smaller sample size from one educational year in one academic medical center in the Netherlands, anonymity of the participants could not be guaranteed. Upon request data could be requested from the Educational Research Review Board of Leiden University Medical Center (secretary: p.g.m.de_jong@lumc.nl) for researchers who meet the criteria for access to confidential data.

**Funding:** The authors received no specific funding for this work.

**Competing interests:** The authors have declared that no competing interests exist.

## Conclusions

Since intrinsic motivation for research, perceptions of research, and curiosity are prerequisites of future research involvement, it seems beneficial to focus on motivation when selecting students for ERPS, allowing students with lower current academic performance to participate in ERPs as well.

## Introduction

Serious concerns have been raised regarding the future of academic medicine in the past decades, as a result of a continued physician-scientist shortage [1–4]. Physician-scientists are health care professionals devoting a substantial amount of their time to both clinical care and research [5, 6]. Thereby, they are pivotal to bridge the gap between science and practice: they are important for both identifying relevant clinical problems to translate into research (i.e. bedside-to-bench) as well as translating research outcomes into clinical practice (i.e. bench-to-bedside) [6–10]. However, the decline in physician-scientists persists and the current workforce is aging [1–4, 10, 11].

A possible solution to retain the physician-scientist workforce could be to engage medical students in research early on in medical training [1, 12–14]. Engagement of undergraduate students in research could help to 1) promote awareness and critical appraisal of research among students, 2) motivate students to conduct research, 3) identify possibilities for a research career among students, and 4) recognize research talent by educators, researchers or physicians [9, 15–17].

The importance of research during medical training has been underlined in many medical educational frameworks and accrediting bodies [18–20]. Furthermore, the Boyer Commission presented a report with ten recommendations for reconstructing undergraduate medical education, with the first recommendation urging to make research-based learning the standard. As a result of the importance given to research by educational frameworks and accrediting bodies, and in line with the Boyer Commission's call to promote engagement of undergraduate students in research, many medical schools started intra- and/or extracurricular initiatives to engage students in research [14, 21, 22]. As involvement in research during medical training is associated with involvement in research during future professional practice, stimulating undergraduate students to engage in research during early phases of medical training could well be seen as a first step in the physician-scientist pipeline [12, 23, 24].

In order to promote in-depth research involvement, an academic mindset, and by extension cultivate future physician-scientists, many medical schools implemented extracurricular research programmes (ERPs) [1, 22]. Such programmes occur under different names and diverse formats, e.g. MD/PhD programmes, capstone programmes, summer research programmes, and Honours programmes [13, 22, 25, 26]. Some previous studies into the effects of ERPs showed that they enhanced students' interest in and appreciation of research, increased research skills and productivity, and promoted continued involvement in research [1, 27–30].

Selection procedures for ERPs, especially Honours programmes, usually focus on grades as a means to select 'excellent' students [31–34]. Ericsson (1993) defines excellence by current performance; perceiving students who have the highest grades as excellent students [35, 36]. When defining excellence in this manner, selection of students for ERPs based on grades seems logical.

However, although high grades may be predictive for knowledge recall, they lack predictive validity for knowledge application and higher order cognitive skills [37]. Conducting research

can be seen as a higher order cognitive skill, which implies that selecting students for ERPs should go beyond just grades. This may certainly be the case in medical Bachelor degree programmes, in which most of the grades are based on exams that focus mainly on knowledge recall. Moreover, the goals of the specific ERPs should be taken into account as well. If the pre-eminent goal of ERPs is to develop an academic mindset and cultivate future physician-scientists, it seems questionable to focus solely on grades. Additionally, it is important to keep in mind that grades do not necessarily reflect all the competencies that are valued in the job market. This might be especially the case for health care professionals who must be able to take on different roles (e.g. communicator, collaborator, health advocate, scholar) [19].

Indeed, an alternative perspective on excellence defines excellent students in terms of *potential* performance, emphasizing the equal importance of motivation next to above-average intellectual ability [38]. This perspective, focusing on motivation as a parameter for excellence as well, might better align with the goal of ERPs to promote research involvement, an academic mindset or even cultivate physician-scientists.

Perceiving students with the highest grades as excellent students, without taking motivation into account when selecting students for ERPs that have the goal to cultivate physician-scientists, might exclude a very important target group of students that are motivated for research and willing to pursue a research-oriented career. In fact, according to Weaver and colleagues, the strongest predictor for a physician-scientist career is indeed an existing passion for research [12]. Previous research also showed that, in line with the Self-Determination Theory, intrinsic motivation for research is related to (further) research involvement in medical school, which in turn is related to research involvement in future professional practice [23, 39]. In addition, in line with the Social Cognitive Theory, research self-efficacy is believed to be related to research motivation and the tendency to conduct research [40]. Furthermore, according to the Theory of Planned Behaviour, perceptions are related to intentions, which in turn are related to the desired behaviour [41]. Lastly, curiosity is identified as an important antecedent for conducting research [23]. In sum, as the main goal of many ERPs is to foster future physician-scientists, intrinsic motivation for research, research self-efficacy beliefs, positive perceptions of research, and curiosity might be valuable objectives to pursue and promote among ERP students.

Students with lower current academic performance might become equally motivated for research as a result of participating within the ERP, thereby contributing to the pool of future physician-scientists. In other words, the question arises if emphasis should shift from grades towards motivation when selecting for ERPs aiming to cultivate future physician-scientists. Without taking motivation into account, an important target group might be excluded.

Therefore, the aim of this study was to investigate if two groups of students in an ERP (students with higher versus lower previous academic performance) differ in subsequent academic performance (i.e. third-year GPA, in-time bachelor completion), ERP performance (i.e. drop-out, number of credits), and motivational factors (i.e. intrinsic motivation for research, research self-efficacy beliefs, perceptions of research, curiosity), by using a prospective, longitudinal approach with a baseline measure. First, we hypothesized that the higher previous performance group will outperform the lower previous performance group on subsequent academic and ERP performance. Second, we hypothesized that the higher previous performance group will not outperform the lower previous performance group on motivational factors because academic skills do not by definition affect motivation for research and students with lower grades may still have a passion for research.

## Materials and methods

### Context

Within the Netherlands, eight universities provide medical education in line with the Dutch National Blueprint for Medical Education, which is based on the CanMEDS [18, 19]. Medical education in all universities is comparable in structure consisting of a six-year undergraduate educational programme, with a three-year programme leading to a Bachelor degree and a subsequent three-year programme leading to a Master degree in Medicine. Leiden University Medical Center (LUMC) is one of eight medical faculties in the Netherlands. Medical students' academic performances are assessed with grades on a 10-point grading scale. Within this grading system, 10 is the highest achievable grade and 6 is the pass grade. GPA is the average of all obtained grades, also reported on a scale of 1 to 10. Grades are not only used to assess students' academic performance, but in general also play a role in selecting students for ERPs. In most ERPs, a GPA of 7 is the threshold for entering. However, on top of the core curriculum, LUMC offers students the possibility to participate in a voluntary ERP (i.e. research-based Honours programme), without requiring a GPA of 7 or higher as the threshold for entering. Every medical student can apply, as in the past years selection was mainly based on self-selection without very strict institutional criteria. Consequently, the way our programme is implemented offers a unique opportunity to compare students with a GPA below and above 7. Students do, however, need to graduate in time and with a GPA of 7 or higher within the regular educational programme in order to receive a certificate from the ERP. Within this programme, LUMC aims to foster research talent and students are provided with the opportunity to conduct research individually. The programme starts in the second year of medical school and lasts two years. On average, about 50–60 motivated students participate in the programme each year, which represents 15–20% of the whole second year cohort of medical students. Furthermore, students need to obtain 30 credits (ECTS, i.e. European Credit Transfer and Accumulation System, which means that students have to invest 28 hours of active study per credit) [26].

### Design and participants

This prospective cohort study is part of a longitudinal study in which one cohort of medical undergraduates is followed through medical education. All students starting medical school in 2016 were asked to participate in the longitudinal study, and requested to fill in one questionnaire each year (e.g. November 2016, January 2018, December 2018). Furthermore, grades and ERP performance characteristics were obtained. In the present study, all students participating in the ERP of the LUMC were included.

### Materials and definitions

To investigate the effect of student group on *academic performance*, GPA of the third year of medical education (GPA3) and time to degree were drawn from university files. To investigate the effect on *ERP performance*, drop-out from and number of ECTS in the programme were drawn from university files as well. Lastly, to examine the effect of student group on *motivational factors*, questionnaire data were used [39]. The questionnaire was based on existing and validated scales, which were adjusted to the medical education setting with a focus on conducting research. Students were asked to score items on a 7-point Likert scale ranging from *1 –'totally disagree'* to *7 –'totally agree'*.

Intrinsic motivation for research was defined as students being motivated to conduct research out of their own interest. The scale consisted of five items (e.g. 'doing research is fun')

based on the Interest/Enjoyment Scale of the Self-Determination Questionnaires [42, 43]. Research self-efficacy was defined as students' beliefs about their ability to conduct research. The scale consisted of three items (e.g. 'I feel I am competent enough to do research') and was self-developed, but inspired by the Dutch General Self-Efficacy Scale and the Academic Efficacy Scale [44, 45]. Perceptions of research were defined as students' beliefs about the value of research. The scale consisted of five items (e.g. 'It is important for medical professionals to have scientific skills') of the Student Perception of Research Integration Questionnaire [46]. Lastly, curiosity was measured with ten items (e.g. 'I enjoy investigating new ideas') of the Epistemic Curiosity Scale [47].

## Procedure

After adjustment of the existing scales, the questionnaire was translated from English to Dutch by using the forward and backward translation procedure. In a pilot study, we pretested the questionnaire amongst ten undergraduate medical students in their second year of medical education, after which two minor adjustment to two items were made. All first-year medical students starting medical training in 2016 were approached by the first author during a workgroup session (T1 baseline—November 2016). The same students were approached again in the first semester of their second (T2—January 2018) and third year (T3—December 2018) of medical school.

Students were informed about the goals and voluntary nature of participating in this study. Additionally, it was explained to students that all data would be used for research purposes and would be processed anonymously. Furthermore, written consent was asked to connect data of all questionnaires and to connect questionnaire data to prior and subsequent academic and ERP performance. This study was approved by the ethical review board of the Netherlands Association of Medical Education: reference number 952.

## Analyses

We used descriptive statistics to report demographics of the participants. We calculated Cronbach's alpha to estimate the reliability of the scales. Mean scores were calculated for the motivational factors. We established normal distribution of the data. Students in the ERP were divided in two groups based on GPA of the first year of undergraduate medical education (GPA1) prior to the start of the ERP: GPA1 $\geq$7 versus GPA1 <7. We used independent t-tests to examine if the two groups differed on the motivational factors at the start of medical training (i.e. T1 –baseline scores). Both univariate as well as multivariate logistic and linear regression analyses were used to compare the two groups of students on academic performance, ERP performance, and motivational outcomes, adjusting for baseline motivation and gender.

To test our first hypothesis that the $\geq$7 group outperforms the <7 group on subsequent academic and ERP performance, we assessed whether the difference in performance was significantly different from zero by looking at 95% confidence intervals. However, to test our second hypothesis that the $\geq$7 group does not outperform the <7 group on motivational outcomes, we need a different approach. More specifically, we sought to demonstrate that there was no difference in motivational outcomes between the two groups. However, testing whether a difference in motivational outcomes is significantly different from zero will not help in demonstrating there is in fact no difference, as lack of statistical significance would not prove *absence of difference* between the two groups. Instead, we assessed whether the difference was not more than a certain pre-set margin. In this so-called non-inferiority design, the non-inferiority margin is the maximum difference below which we consider the groups to be not meaningfully different [48]. We elaborated on the non-inferiority margin and reached consensus on a non-

inferiority margin of 0.5 on the motivational scales. Hence, we tested if the groups differ by less than 0.5 on motivational outcomes, and thus that the difference is significantly smaller than 0.5, by assessing whether 0.5 is outside the 95% confidence interval. If so, we can conclude that the <7 group and the ≥7 group do not meaningfully differ in motivational outcomes, and the <7 group at most performs only marginally worse. We analysed all data using IBM SPSS Statistics version 23.

## Results

Within this cohort existing out of 315 students, a total of 59 students participated in the ERP. All 59 students consented to participate in the current study, of whom 13 (22%) were male and 46 (88%) were female students. This male/female distribution is comparable to the distribution within the whole cohort of medical students. The 59 students were divided in 29 students in the ≥7 group (49.2%) and 30 students (50.8%) in the <7 group. Baseline scores of the two groups on GPA1, intrinsic motivation for research, research self-efficacy, perceptions of research, and curiosity can be found in Table 1. All students were included in the analyses of academic and ERP performance. In total, 57 out of 59 ERP students participated in the baseline survey (96.6%), and 54 out of 59 students participated in the third-year survey (91.5%) addressing the motivational factors. Cronbach's alpha was calculated for the scales of the questionnaire and ranged from .81 to .88 at baseline T1 (November 2016, first year of medical school) and from .80 to .89 at T3 (December 2018, third year of medical school). At baseline, the two groups of students did not differ significantly on the motivational factors. GPA3, in-time bachelor completion, drop-out rates, number of credits within the programme and T3 motivational scores are reported in Table 2.

**Table 1. Baseline characteristics divided by ≥7 (n = 29) and <7 (n = 30) student group.**

| | *≥7 student group* | *<7 student group* |
|---|---|---|
| Gender | | |
| Male | 6 (20.7%) | 7 (23.3%) |
| Female | 23 (79.3%) | 23 (76.7%) |
| GPA year 1 | | |
| n | 29 | 30 |
| M (SD) | 7.64 (.44) | 6.73 (.15) |
| Min-Max | 7.02–8.92 | 6.41–6.96 |
| Intrinsic motivation T1 | | |
| n | 28 | 29 |
| M (SD) | 5.98 (.64) | 5.72 (.71) |
| Min-Max | 4.8–7.0 | 4.2–6.8 |
| Research self-efficacy T1 | | |
| n | 28 | 29 |
| M (SD) | 5.13 (.98) | 5.14 (.96) |
| Min-Max | 3.0–7.0 | 3.0–6.7 |
| Perceptions of research T1 | | |
| n | 28 | 29 |
| M (SD) | 5.92 (.73) | 5.81 (.84) |
| Min-Max | 3.8–7.0 | 2.8–6.8 |
| Curiosity T1 | | |
| n | 28 | 29 |
| M (SD) | 5.45 (.65) | 5.47 (.73) |
| Min-Max | 4.1–6.6 | 4.3–6.9 |

**Table 2. Overview of outcome measures divided by ≥7 (n = 29) and <7 (n = 30) student group.**

|  | *≥7 student group* | *<7 student group* |
|---|---|---|
| GPA year 3 |  |  |
| M (SD) | 7.62 (.41) | 7.14 (.25) |
| Min-Max | 6.77–8.65 | 6.70–7.75 |
| In-time bachelor completion |  |  |
| no (%) | 4 (13.8%) | 5 (16.7%) |
| yes (%) | 25 (86.2%) | 25 (83.3%) |
| ERP drop-out |  |  |
| no (%) | 18 (62.1%) | 9 (30%) |
| yes (%) | 11 (37.9%) | 21 (70%) |
| Amount of ECTS |  |  |
| M (SD) | 23.72 (17.39) | 12.17 (13.29) |
| Min-Max | 0–55 | 0–41 |
| Intrinsic motivation T3 |  |  |
| n | 26 | 28 |
| M (SD) | 5.98 (.61) | 5.81 (.69) |
| Min-Max | 4.8–7.0 | 4.0–7.0 |
| Research self-efficacy T3 |  |  |
| n | 26 | 28 |
| M (SD) | 5.10 (.98) | 4.70 (.81) |
| Min-Max | 3.3–7.0 | 2.0–6.0 |
| Perceptions of research T3 |  |  |
| n | 26 | 28 |
| M (SD) | 5.45 (.79) | 5.75 (.87) |
| Min-Max | 4.0–7.0 | 3.8–7.0 |
| Curiosity T3 |  |  |
| n | 26 | 28 |
| M (SD) | 5.29 (.64) | 5.49 (.69) |
| Min-Max | 3.8–6.7 | 4.1–6.9 |

## Academic performance

An effect of student group on GPA3 was found, with students in the <7 group performing significantly lower in the third year of medical education ($\beta$ = -.48, 95%CI = -.66 - -.29). This effect remained after adjusting for gender and the motivational factors at baseline ($\beta$ = -.46, 95%CI = -.67- -.25). However, there was no effect on in-time bachelor completion (crude OR = .80, 95% CI = .19–3.33; adjusted OR = .83, 95%CI = .17–4.00).

## ERP performance

With regard to ERP performance, a significant effect was found from student group on ERP drop-out. The odds for ERP drop-out were significantly higher in the <7 group (OR = 3.82, 95% CI = 1.29–11.28), also after adjusting for gender and motivational baseline scores (OR = 4.25, 95% CI = 1.29–13.94). Furthermore, a significant effect was found regarding the number of credits in the programme, with students in the <7 group obtaining less credits than students in the ≥7 group (crude $\beta$ = -11.55, 95%CI = -19.60 - -3.50; adjusted $\beta$ = -12.52, 95%CI = -20.83 - -4.20).

## Motivational factors

With regard to the motivational factors, the non-inferiority margin was set at 0.5 points. Our findings showed that, after adjusting for gender and the motivational factors at baseline, -0.5

**Table 3. Regression model of the effect of type of student on the motivational factors in the third year of medical education.**

|  | Crude | Adjusted for gender and motivational baseline scores |
|---|---|---|
|  | β (95%CI) | β (95%CI) |
| Intrinsic Motivation | -.16 (-.52 - .19) | -.13 (-.44 - .19) |
| Research self-efficacy | -.40 (-.89 - .09) | -.40 (-.89 - .09) |
| Perceptions of research | .30 (-.16 - .75) | .29 (-.16 - .74) |
| Curiosity | .20 (-.16 - .56) | .12 (-.21 - .44) |

*reference: ≥7 group

was not in the 95% confidence interval for intrinsic motivation for research (β = -.13, 95%CI = -.44 - .19), perceptions of research (β = .29, 95%CI = -.16 - .74), and curiosity (β = .12, 95%CI = -.21 - .44). Thus, with 95% confidence, the difference in these motivational factors is smaller than 0.5 and students in the <7 group were not inferior to ≥7 group when it comes to these motivational factors. When looking at research self-efficacy, -0.5 is within the confidence interval (β = -.40, 95%CI = -.89 - .09). Therefore, we cannot conclude that the <7 group was not inferior to the ≥7 group. An overview of these findings can be found in Table 3.

## Discussion

Within this study, we compared two groups of students on three outcome levels: academic performance, ERP performance, and motivational factors. We hypothesized that the ≥7 group would outperform the <7 group on academic and ERP performance, but not on motivational factors. In line with our first hypothesis, the <7 group had lower academic performance (GPA3), significantly higher odds for ERP drop-out and less credits within the ERP compared to the ≥7 group. Confirming our second hypothesis, the <7 group is not inferior to the ≥7 group on intrinsic motivation for research, perceptions of research, and curiosity in the third year of medical education. In other words, intrinsic motivation for research, perceptions of research, and curiosity in the third year of medical education did not differ meaningfully between both groups. The only contradiction to our hypotheses was found on in-time bachelor completion, as the two groups of students did not significantly differ in obtaining their bachelor degree in the appointed amount of time.

In line with our hypotheses, the <7 group obtained lower levels in GPA3, but did however not seem to differ in time to obtain their degree. ERPs expose students to additional workload on top of their regular medical training [34, 49]. One concern when it comes to including students beyond the 'excellent' and 'high-achieving' student population in certain programmes, is that these students might not be able to combine the additional workload with the regular courses, and that ERP participation will have a negative impact on academic performance in the regular programme. As we did not find a significant difference in in-time bachelor completion, it could be that the <7 group, though on average obtaining 0.5 points lower with regard to GPA3, did not differ from the ≥7 group on in-time bachelor completion. Moreover, the mean difference between the two groups on GPA3 narrowed as compared to GPA in the first year of medical school, as the GPA of the <7 group increased to a larger extent. These findings indicate that ERP participation is not at the expense of the regular programme. ERP participation might even lead to a greater advantage for the <7 group, as participating in the programme may even enhance their GPA in the subsequent years. An explanation for this could be that students in the ERP are surrounded by highly motivated peers [49]. This is in line with the 'reflected glory effect', referring to the tendency individuals have to relate one's self-

perceived ability to the success of others [50]. Within this context, students in the <7 group might identify themselves with the selective group of high-achieving and motivated peers, which has a positive impact on their self-perceived ability. From this perspective, the presence of the ≥7 group is a positive factor for the performance of the <7 group. This, in turn, is in line with findings by another study showing that improved self-concept is related to increased learning outcomes [51].

Our findings suggest that the <7 student group obtained significantly less credits within the programme. This is probably associated with the fact that for students in the <7 group the odds for ERP drop-out were about four times as high. It is remarkable that, though comparable in motivation for research, the drop-out in the <7 group is higher as compared to the ≥7 group, possibly implying that motivation does not lead to ERP completion. The question arises if attrition from the ERP results from a lack of ability to conduct research, or that other reasons might lead to the decision to quit the programme among students. Dropping out of the programme might not by definition mean that students are deterred from research. A reason for the higher ERP drop-out rate may be that, although students with GPA <7 are allowed to participate within the programme, a requirement is that students graduate and receive their medical Bachelor's degree with an average grade of 7 or higher to obtain the ERP certificate [26]. Students who already started the programme with lower grades might feel they will not meet this requirement, and could therefore decide to quit the programme in advance. In addition, this rule sends out the implicit message that students scoring below 7 are not the type of students that are supposed to be enrolled within such ERPs [52]. Some students within the regular cohort also voluntarily conduct extracurricular research without following the structured ERP, so it could be that students dropping out from the ERP decide to follow this path as well. Another possible explanation for the higher chance of dropping out in the <7 group may be the 'big fish little pond–effect' [50], which has the opposite effect for other types of students as compared to the reflected glory hypothesis. According to the big fish little pond–effect, students' self-perceived ability is determined by the comparison with peers. Students participating in an ERP compare themselves with the smaller group of participants within the programme, while largely surrounded by high-achieving, 'top of their class' peers. A similar student in the regular programme will compare itself with the bigger pond of students, differing in cognitive ability. As a result of this change in reference, the <7 group within the programme might have lower levels of self-perceived ability because they are surrounded by some 'big fish' (i.e. the high-achieving GPA1 ≥7 students) in their little pond (i.e. smaller group of programme participants) [50, 53, 54].

Should this higher level of drop-out, then, be a reason to only include 'excellent' students in certain programmes in the future? From an efficiency perspective, one might say that drop-out within a rather costly programme should be avoided. Furthermore, it could be argued that select groups of students receive additional education which needs to be justified, especially because these graduates are more appealing for job recruiters [53]. Lastly, one might wonder if students not completing the ERP will be able to deal with the pressures in future professional practice, for instance combining research and clinical duties. These perspectives might contribute to the idea of solely including high-achieving students in such extracurricular programmes to prevent attrition, aligned with Ericsson's perspective on 'excellence'.

However, when evaluating ERPs, a focus on academic and ERP performance might provide an incomplete image when aiming to deliver professionals who fit the needs of the specific field [33, 53]. When looking at the *motivational factors* in the current study, our results are inconclusive with respect to research self-efficacy beliefs in the third year of medical education, although they seem to be somewhat higher in the ≥7 group. A study by Kool and colleagues [55] showed that high-achieving students were more performance oriented, defined as

students' pursuit to outperform peers and show their own abilities, which might explain the higher levels of confidence in their own abilities among the students in the ≥7 group. In addition, for some students the big fish little pond–effect might apply here as well. A practical implication derived from these findings might be to support motivated, above-average ability students in ERPs in such ways that their research self-efficacy beliefs are enhanced, as research self-efficacy is related to research motivation and the tendency to conduct research [40].

More importantly, our study showed that the two groups of students are comparable in intrinsic motivation for research, perceptions of research, and curiosity in the third year of medical education. In line with major theories like Self-Determination Theory, Social Cognitive Theory, and Theory of Planned Behaviour, as well as findings from previous studies, these constructs are related to actual research engagement [23, 39–43].

Thus, despite the higher ERP drop-out rates in the <7 group, the group is not inferior to the ≥7 group on the desired outcomes that are imperative to cultivate future physician-scientists. In addition, one might say these students are supported in their development regardless of whether they eventually finished the ERP. But above all, if the pre-eminent goal of ERPs is to develop future physician-scientists, this goal is not endangered by including students with lower academic performances in their first year of medical training. In fact, these students might well belong in the target group when aiming to cultivate future physician-scientists and selection based solely on grades poses the risk to exclude a motivated group of students from the physician-scientist training pipeline.

To summarize, students in the <7 group quit the ERP more often and have lower GPA in the third year of medical education, but ERP participation may help to enhance student GPA of the first years of undergraduate medical study in the <7 group. More importantly, the <7 group scored comparable to the ≥7 group on intrinsic motivation for research, perceptions of research and curiosity, which are all motivational factors underlying research involvement in future professional practice. Therefore, when aiming to cultivate future physician-scientists, our findings imply that the perspective on excellence emphasizing *potential performance* and the equal importance of motivation is more aligned with the aims of ERPs. Especially when taking into account that medical students invest a great amount of academic effort before entering medical school and are selected on, among others, cognitive abilities [56]. To conclude, this could mean that, in order to use ERPs as a step in the physician-scientist pipeline, motivation should be given importance in selecting students for ERPs, allowing students with a GPA lower than seven to participate within such programmes as well. This could be established by using a selection model in which GPA is not perceived as a threshold to enter the ERP. Insights in motivation of students willing to self-select within the ERP could be elucidated by, for instance, asking students to write a motivation letter to reflect on their feelings of competence within the regular educational programme (i.e. academic self-efficacy) and their motivation to participate within the specific ERP. Furthermore, it could be valuable to offer students a low-threshold activity in which they can get acquainted with the ERP to substantiate their willingness to participate within the ERP.

## Strengths, limitations and future research

First, our study was performed within a single institute. However, our medical curriculum is comparable to other medical curricula as the educational programme is based on the Dutch National Blueprint for Medical Education, which in turn is aligned with, among others, the CanMEDS [18, 19]. Additionally, many medical schools worldwide provide undergraduate students with ERPs. Second, the outcome measures of the current study are not long-term measures like, for instance, publication rates. However, previous studies have shown that both

research involvement within medical training, as well as the measured motivational factors within this study, are related to long-term scientific involvement [23, 24, 39]. Valuable for future research might be to include long-term effects with scholarly output (e.g. publications and conference contributions), and a career as a physician-scientist. Third, the groups within our study were relatively small. Therefore, an interesting future research avenue might be to conduct this study within different contexts and, when possible, larger groups. Furthermore, future research could focus on identifying causes of lower credits in the <7 group, as well as investigating if ERP drop-outs perceive this negatively and what is needed to support all different types of students and promote every student to flourish within such ERPs. A qualitative approach could help to provide more in-depth insights into the above mentioned topics. In addition, reasons for drop-out and subsequent intentions to pursue research, or lack thereof, might be valuable to identify.

## Conclusion

Two groups of students within an ERP were compared on three outcome levels: academic performance, ERP performance, and motivational factors. The <7 group obtained lower levels of GPA3 and had significantly higher odds for ERP drop-out. On the contrary, the <7 group did not differ from the ≥7 group on in-time bachelor completion, and had comparable levels of intrinsic motivation for research, perceptions of research, and curiosity in the third year of medical education, which are all factors underlying research involvement in future professional practice. Therefore, for ERPs aiming to develop future physician-scientists, a shift from an emphasis solely on grades towards taking motivation into account could be beneficial for the selection for such programmes, allowing students with lower current performance to participate as well.

## Author Contributions

**Conceptualization:** Belinda W. C. Ommering, Floris M. Van Blankenstein, Ada Kool, Friedo W. Dekker.

**Formal analysis:** Belinda W. C. Ommering, Friedo W. Dekker.

**Investigation:** Belinda W. C. Ommering, Friedo W. Dekker.

**Methodology:** Belinda W. C. Ommering, Floris M. Van Blankenstein, Merel van Diepen, Friedo W. Dekker.

**Resources:** Nelleke A. Gruis.

**Software:** Merel van Diepen.

**Supervision:** Floris M. Van Blankenstein, Merel van Diepen, Friedo W. Dekker.

**Writing – original draft:** Belinda W. C. Ommering.

**Writing – review & editing:** Belinda W. C. Ommering, Floris M. Van Blankenstein, Merel van Diepen, Nelleke A. Gruis, Ada Kool, Friedo W. Dekker.

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
