## [Decision Letter · Decision Letter 0]

30 Apr 2021

PONE-D-21-11278

The importance of motivation in selecting undergraduate medical students for extracurricular research programmes

PLOS ONE

Dear Dr. Ommering,

Thank you for submitting your manuscript to PLOS ONE. After careful consideration, we feel that it has merit but does not fully meet PLOS ONE’s publication criteria as it currently stands. Therefore, we invite you to submit a revised version of the manuscript that addresses the points raised during the review process.

Kindly find the reviewers' comments. My personal reading of the manuscript shows it is well-written and after the required changes can be an important contribution to the journal. 

We look forward to receiving your revised manuscript.

Kind regards,

Pathiyil Ravi Shankar

Academic Editor

PLOS ONE

Additional Editor Comments (if provided):

My commenst are: 

Lines 80 and 81: The authors should describe the recommendations of the Boyer Commission for the benefit of readers.

How are students selected currently for the ERPs? Is only the GPA considered? How are students matched to different ERPs?

How exactly do the authors propose to measure motivation for research and other characteristics while selecting students for ERP? What model of student selection are they proposing and what weight will be provided to GPA and other factors including motivation?  

The authors should also make available/deposit the data associated with the research in a publicly accessible data repository. Personal details can be deidentified if needed. 

Journal Requirements:

2) Please provide additional details regarding participant consent. In the ethics statement in the Methods and online submission information, please ensure that you have specified what type you obtained (for instance, written or verbal, and if verbal, how it was documented and witnessed). If your study included minors, state whether you obtained consent from parents or guardians. If the need for consent was waived by the ethics committee, please include this information.

Reviewers' comments:

Reviewer's Responses to Questions

**Comments to the Author**

1. Is the manuscript technically sound, and do the data support the conclusions?

Reviewer #1: Yes

Reviewer #2: Yes

2. Has the statistical analysis been performed appropriately and rigorously? 

Reviewer #1: Yes

Reviewer #2: Yes

3. Have the authors made all data underlying the findings in their manuscript fully available?

Reviewer #1: Yes

Reviewer #2: Yes

4. Is the manuscript presented in an intelligible fashion and written in standard English?

Reviewer #1: Yes

Reviewer #2: Yes

5. Review Comments to the Author

Reviewer #1: This is a well written manuscript. The data is interesting. It is a prospective cohort study with undergraduate medical students who filled in a yearly questionnaire on motivational factors.

Two medical student groups participating in an ERP: students with first-year grade point average (GPA) ≥7 versus <7 on a 10- point grading scale. 70% students in the GPA <7 group dropped out of ERP compared to 38% in the GPA >7 group. The lower GPA group loss had lower GPA at the end of the study.

In terms of questionnaires related intrinsic motivation for research, perceptions of research, and curiosity at the end of the study there was no difference between groups. The authors propose allowing students with lower current academic performance to participate in ERPs is beneficial.

The conclusion does not seem supported by the results. Despite answering the questions related to research motivation and curiosity similarly, the majority of the students in the lower GPA group did not complete the ERP program. The research curiosity and motivation did not translate into continuation / completion of ERP.

How did the select students who continued the ERP (9/30) in the lower GPA group score on the research perception questionnaires compared the students who continued the ERP in the higher GPA group?

It would be interesting to follow this cohort to see how many will become actual physician scientists.

Table 1 and 2: please insert a column for P value between the groups.

Reviewer #2: The hypothesis is relevant and the methodology is aligned with the objectives. Qualitative data gathering with factor analysis for the conclusions derived would have strengthened the conclusions of the study. The study would be more meaningful if an attempt was made to identify causes of lesser credits in <7 group inspite of comparable intrinsic motivation. If the students in <7 group dropped out to meet academic requirements , it would establish the counter argument that intrinsic motivation alone is not enough for participation in ERP. Authors could highlight regarding remedial measures offered, if any for the students in <7 group to clarify its role in lesser in group difference in GPA3 as compared to GPA 1.

6. PLOS authors have the option to publish the peer review history of their article (what does this mean?). If published, this will include your full peer review and any attached files.

Reviewer #1: No

Reviewer #2: **Yes: **Vinutha Shankar MS

---

## [Author Response · Author response to Decision Letter 0]

11 Jun 2021

Dear Pathiyil Ravi Shankar,

We would like to thank you for the opportunity to revise our manuscript. In addition, we would like to thank you and the reviewers for the helpful review comments, which are greatly appreciated. The comments offer us guidance in the review process and help to improve our manuscript. Below we provide an overview on how we processed the review comments. 

Editor comments

“Lines 80 and 81: The authors should describe the recommendations of the Boyer Commission for the benefit of readers.”

Done as requested (see revised manuscript, page 4; lines 80-84).

“How are students selected currently for the ERPs? Is only the GPA considered? How are students matched to different ERPs?”

Our study is conducted within Leiden University Medical Center (LUMC), which offers one extracurricular research programme for which every student can apply. That means that self-selection is the basis for enrolment within the single programme that the LUMC offers. There are no institutional selection criteria, so GPA is not considered to be a factor to deny students to participate within the programme. Students do, however, need to graduate in time and with a GPA of 7 or higher within the regular educational programme in order to receive a certificate from the ERP. This means that GPA is not considered as a selection requirement when entering the ERP, but within our current ERP it does play a role in order to receive a certificate. The self-selection approach to enter the ERP is not very common, as most ERPs select students based on GPA and regard a GPA of at least 7 (on a 10-point grading scale) to be sufficient to participate in an ERP. However, as the programme within our institute is accessible for every student, this did gave us the unique opportunity to compare students scoring above the 7 and below the 7, the latter being a motivated group of students that might want to pursue a research oriented career but in many other institutes and programmes would not have been given the chance to participate and act on their intrinsic motivation for research. In order to make the distinction between how students are generally selected for ERPs and our approach to letting students enrol by self-selection more apparent, within the revised manuscript on page 8 (line 179) we now explicitly mention that there is no GPA threshold for entering the ERP. Furthermore we structured the paragraph different within the revised manuscript, in such a way that first the general context of ERP selection is described, after which the way self-selection is handled within the LUMC immediately follows while also mentioning the requirements to obtain the ERP certificate (page 8, lines 177-184). We hope this clarifies the matter. 

“How exactly do the authors propose to measure motivation for research and other characteristics while selecting students for ERP? What model of student selection are they proposing and what weight will be provided to GPA and other factors including motivation?” 

In line with our vision, we strive towards a selection model in which GPA is not perceived as a threshold to enter the ERP. As previous studies showed that intrinsic motivation for research is related to actual research involvement in future professional practice, combined with studies showing that grades are not predictive for higher order cognitive skills, we might exclude potential future physician-scientists when denying motivated students access to the ERP just because their GPA is below 7. Our key message therefore is to offer motivated students the opportunity to participate within the ERP regardless of their academic GPA when starting the ERP, because these motivated students might be a target group when aiming to foster research talent and cultivate future physician-scientists (see page 6 and 7 of the manuscript). The finding that the <7 group is not inferior to the ≥7 group within the ERP corroborates this vision. We hope this clarifies that we do not propose to give a certain amount of weight to GPA and other factors, we believe in the self-selection model enabling students that are passionate about research and willing to enter or pursue in this pipeline. Within the revised manuscript, we now elaborate on how to obtain insights in motivation for research (page 21, line 449-456). We feel that students, for instance, could be asked to write a motivation letter reflecting on their feelings of competence within the regular educational programme (i.e. academic self-efficacy) and their motivation to participate within the specific ERP. Furthermore, we believe it could be valuable to offer students a low-threshold activity in which they can get acquainted with the ERP to substantiate their willingness to participate within the ERP. It could also be interesting, however, to measure motivation for research while selecting students for the ERP, for which we did develop a questionnaire. This could, for instance, help to provide insights into the development of motivation for research through the ERP. 

“The authors should also make available/deposit the data associated with the research in a publicly accessible data repository. Personal details can be deidentified if needed.” 

We agree that from an open access perspective data should be made available when possible. However, in the consent form, we informed the participating students that data would not be traceable to individual participants. As this is a somewhat smaller sample size from one educational year in one academic medical center in the Netherlands, anonymity of the participants could not be guaranteed when the data associated with the research is deposited in a publicly accessible data repository. We hope that the academic editor acknowledges that due to these ethical reasons we are not able to deposit are data. Of course, when approached by other researchers, we would be open and happy to collaborate with those interested. 

Journal requirements

“Please ensure that your manuscript meets PLOS ONE's style requirements, including those for file naming.”

We thoroughly looked at the Plos One submission guidelines and style templates to ensure that our manuscript meets the style requirements, which led to one minor adjustment on page 8 (line 164) of the manuscript: we replaced the level 1 heading ‘Methods’ by ‘Materials and methods’. 

“Please provide additional details regarding participant consent. In the ethics statement in the Methods and online submission information, please ensure that you have specified what type you obtained (for instance, written or verbal, and if verbal, how it was documented and witnessed). If your study included minors, state whether you obtained consent from parents or guardians. If the need for consent was waived by the ethics committee, please include this information.”

In the procedure-subheading at page 11 we already mentioned that students’ consent was asked to connect data of all questionnaires and to connect questionnaire data to prior and subsequent academic and ERP performance. However, within the revised manuscript we now explicitly mentioned that written consent was asked (see page 11, line 241 of the revised manuscript). 

Reviewer 1

“This is a well written manuscript. The data is interesting. It is a prospective cohort study with undergraduate medical students who filled in a yearly questionnaire on motivational factors. Two medical student groups participating in an ERP: students with first-year grade point average (GPA) ≥7 versus <7 on a 10- point grading scale. 70% students in the GPA <7 group dropped out of ERP compared to 38% in the GPA >7 group. The lower GPA group loss had lower GPA at the end of the study. In terms of questionnaires related intrinsic motivation for research, perceptions of research, and curiosity at the end of the study there was no difference between groups. The authors propose allowing students with lower current academic performance to participate in ERPs is beneficial. The conclusion does not seem supported by the results. Despite answering the questions related to research motivation and curiosity similarly, the majority of the students in the lower GPA group did not complete the ERP program. The research curiosity and motivation did not translate into continuation / completion of ERP.”

Thank you for the compliments regarding our manuscript and for summarizing our findings correctly. Indeed, the drop-out rate is higher in the <7 group and their GPA is lower in year three as compared to the ≥7 group. However, the <7 group had lower levels of GPA to begin with and when looking at Table 1 and 2, it seems that their GPA increased to a larger extent then the GPA in the ≥7 group. One of the mentioned concerns when considering to let students below a certain threshold GPA apply and participate within an ERP is that it will negatively impact academic performance within the regular educational programme. Our findings imply that this is not the case and that ERP participation is not at the expense of academic performance within the regular programme for the students starting the programme with a GPA below 7. It might even lead to a greater advantage as the mean difference between the two groups decreases because of the greater increase in GPA in the <7 group (for our interpretation and explanation through the lens of the “reflected glory” effect, we kindly refer to page 17, lines 359-366 of our manuscript). We apologize for the confusion and within the revised manuscript we now explicitly mentioned that the GPA of the <7 group increased to a larger extent (page 17, lines 355-356). We hope this helps to clarify our message. 

The reviewer comes with an interesting interpretation by stating that the majority of the students within the <7 group did not finish the programme, and that research curiosity and motivation did not translate into ERP completion. We agree with the reviewer that this could indeed be the case, however, we believe that the academic requirements held by our institution (i.e. students must graduate and receive their medical Bachelor’s degree with an average grade of 7 or higher to obtain the ERP certificate) result in students choosing to quit the programme in advance, while still being the curious and motivated student that started the programme. Especially, as some students communicate these type of reasons to the ERP-coordinator. In the revised manuscript we have also added the notion as mentioned by the reviewer that it could mean that curiosity and motivation alone is not enough (see page 18, lines 369-371 of the revised manuscript). We hope our future research will shed light on this particular issue. To conclude, we do feel the need to mention here that motivation on the long run feels more important than whether or not the student obtained the certificate. One third of the students in the <7 group did obtain their certificate and otherwise we would have missed out on this group. We do however plan to study in more detail why students drop out of the programme and how their motivation for research could be described after dropping out. 

“How did the select students who continued the ERP (9/30) in the lower GPA group score on the research perception questionnaires compared the students who continued the ERP in the higher GPA group?”

When comparing these 18 (≥7 group) versus 9 (<7 group) students, our findings show that the main conclusions to be drawn from the results are comparable. When comparing these smaller groups our findings showed that, after adjusting for gender and the motivational factors at baseline, -0.5 was not in the 95% confidence interval for intrinsic motivation for research (β = .10, 95%CI = -.42 - .63), perceptions of research (β = .65, 95%CI = -.11 – 1.41), and curiosity (β = .24, 95%CI = -.44 - .93). Thus, with 95% confidence, the difference in these motivational factors is smaller than 0.5 and students in the <7 group are not inferior to ≥7 group when it comes to these motivational factors. When looking at research self-efficacy, -0.5 is within the confidence interval (β = -.58, 95%CI = -1.32 - .17). Therefore, we cannot conclude that the <7 group is not inferior to the ≥7 group. However, the groups within this analysis were very small so caution is needed when interpreting these results. 

“It would be interesting to follow this cohort to see how many will become actual physician scientists.”

We absolutely agree with the reviewer that it would be interesting to follow this cohort to see how many students will become actual physician-scientists. Within our future research paragraph on page 22 (line 468) of the revised manuscript we have added ‘a career as a physician-scientist’ (next to scholarly output like publications and conference contributions) as a possible long-term effect that would be interesting for future research. 

“Table 1 and 2: please insert a column for P value between the groups.”

Table 1 illustrates the descriptives of the participants within our group – as strictly speaking no formal hypothesis is tested here, we deliberately did not insert the p value. We believe that inserting the p value (or confidence intervals) would be informative when a research question is formed and consequently a hypothesis is tested, however when reporting the background or baseline characteristics of the participants, we feel it is not necessary. We hope that, after elaborating on our intention or aim with Table 1, we have clarified why the p value is not mentioned here. With regards to Table 2 we do not have a problem with reporting significance. We actually did report significance, however, at a different place within the manuscript: confidence intervals are reported in the main text as well as in Table 3 of our manuscript. Our intention was to keep Table 2 easy accessible and we aimed at easy interpretation of the table, reporting significance here would, in our opinion, decrease readability (as for instance both linear and logistics regressions are used). Therefore, we chose to report confidence intervals in the main text for all outcome measures, and we also reported the outcomes regarding the motivational factors (all tested with linear regression analyses with a non-inferiority approach) in Table 3. We hope this clarifies why we have made the current decision as well.

Reviewer 2

“The hypothesis is relevant and the methodology is aligned with the objectives. Qualitative data gathering with factor analysis for the conclusions derived would have strengthened the conclusions of the study. The study would be more meaningful if an attempt was made to identify causes of lesser credits in <7 group inspite of comparable intrinsic motivation. If the students in <7 group dropped out to meet academic requirements , it would establish the counter argument that intrinsic motivation alone is not enough for participation in ERP. Authors could highlight regarding remedial measures offered, if any for the students in <7 group to clarify its role in lesser in group difference in GPA3 as compared to GPA 1.”

We agree with the reviewer that a qualitative approach would be an interesting next step to further investigate the importance of motivation for research in selecting students for an extracurricular research programme (ERP), which is why we have added this as a future research avenue within the revised manuscript on page 22 (lines 474-475). We appreciate the interesting perspective mentioned by the reviewer to identify causes of lesser credits in the <7 group, and see this as a valuable direction for future research as well. Within our discussion section, we connected the lower credits to the drop-out rate. However, other factors might play a role as well which is why we have added this suggestion of the reviewer to our future research paragraph as well (see page 22, lines 471-472 of the revised manuscript). With regard to the reasons for drop-out, we indeed mentioned the possibility that students dropped out because of the academic requirements that need to be obtained in order to receive the certificate at the end of the extracurricular research programme. Every student can enrol within the ERP in our institute, the programme does not exclude students based on their first-year GPA. However, though this could be deemed as controversial, the institute does demand that students’ GPA for the regular medical educational programme is above 7 (on a 10-point grading scale) at the end of the ERP in order to receive the certificate. This not only sends out the implicit message that students scoring below 7 are not the type of students that are supposed to be enrolled within such ERPs (see manuscript page 18, lines 374-384), but it could also be demotivating and possibly decreases students initial intrinsic motivation for research. Especially when considered that this requirement, according to Self-Determination Theory, could well be an initiator of extrinsic motivation for research. However, it could also be that students indeed drop-out because of the academic requirement, but that they are still intrinsically motivated to conduct research as also mentioned on page 18 of the revised manuscript. With the current rules as they are, our findings could indeed be interpreted in such a way that intrinsic motivation alone is not enough to participate in the ERP, as also mentioned in our response to the first reviewer and now added within the revised manuscript on page 18 (lines 369-371). However, we feel that the academic requirement for a GPA of >7 at the end of the third year of medical training limits students that were initially intrinsically motivated for research to succeed within the programme and their journey towards a research oriented career. We hope that future research will shed light on actual reasons for drop-out, as it would be very interesting to gain in-depth knowledge on why some students persist within the programme while others drop-out, while both type of students might be the students that should be targeted for future physician-scientist careers. To conclude, no remedial measures were offered for the students in the <7 group. 

Again, we would like to express our gratitude to you and the reviewers for the detailed and constructive comments. We are very thankful for the time invested in our manuscript. We really feel it helped us to improve our manuscript considerably, and hope you acknowledge our improvements.

If any questions or concerns remain, please do not hesitate to contact us. Thank you for receiving our revised manuscript, we look forward to our collaboration on this manuscript.

On behalf of all authors,

Kind regards,

Belinda Ommering

---

## [Decision Letter · Decision Letter 1]

16 Aug 2021

PONE-D-21-11278R1

The importance of motivation in selecting undergraduate medical students for extracurricular research programmes

PLOS ONE

Dear Dr. Ommering,

Thank you for submitting your manuscript to PLOS ONE. After careful consideration, we feel that it has merit but does not fully meet PLOS ONE’s publication criteria as it currently stands. Therefore, we invite you to submit a revised version of the manuscript that addresses the points raised during the review process.

Kindly revise as per the comments of the reviewer of this version of the manuscript to be considered further. 

We look forward to receiving your revised manuscript.

Kind regards,

Pathiyil Ravi Shankar

Academic Editor

PLOS ONE

Journal Requirements:

Additional Editor Comments (if provided):

Reviewers' comments:

Reviewer's Responses to Questions

**Comments to the Author**

1. If the authors have adequately addressed your comments raised in a previous round of review and you feel that this manuscript is now acceptable for publication, you may indicate that here to bypass the “Comments to the Author” section, enter your conflict of interest statement in the “Confidential to Editor” section, and submit your "Accept" recommendation.

Reviewer #2: All comments have been addressed

Reviewer #3: (No Response)

2. Is the manuscript technically sound, and do the data support the conclusions?

Reviewer #2: Yes

Reviewer #3: Partly

3. Has the statistical analysis been performed appropriately and rigorously? 

Reviewer #2: Yes

Reviewer #3: Yes

4. Have the authors made all data underlying the findings in their manuscript fully available?

Reviewer #2: Yes

Reviewer #3: Yes

5. Is the manuscript presented in an intelligible fashion and written in standard English?

Reviewer #2: Yes

Reviewer #3: Yes

6. Review Comments to the Author

Reviewer #2: The comments have been addressed and the project has to be taken forward as per suggestions to be more useful and relevant.

Reviewer #3: Reviewer 1 comments:

The importance of motivation in selecting undergraduate medical students for extracurricular research programmes (Manuscript Number PONE-D-21-11278R1)

This study attempts to throw light on the common oversight of an important benchmark in performance evaluation of medical researchers.

The paper is detailed, well explained and has incorporated good statistical tools. There a few observations made by the reviewer:

1. Kindly correct grammar inconsistencies at one or two areas in the paper (Eg. Inconsistencies with tenses: Line 273 – ‘All students were included’ is more appropriate as the study has been reported in the past tense throughout the paper)

2. If ERP drop-outs were more in the <7 group and this was statistically significant, one can argue that lack of motivation played an important role in the discontinuation from the ERP. Rather than dogmatically defending the findings with two contradictory theories for comparison of two different parameters between the same two study groups (viz. Reflected glory effect and Big fish in small pond theory), the acceptance to the paper can be enhanced with an open-mind approach that is:

- Authors can state in the paper that ‘the higher drop-out rate in <7 group could be justified by the big fish in small pond theory or could be attributed to the lack of motivation in the <7 group. Although the questionnaire shows no difference in motivation between the two groups, the finding of increased drop-outs confounds the result and warrants further research to investigate the reason for increased drop-outs in the <7 GPA group’

- If authors discuss that “Reflected glory effect” explains the increased self-perceived ability of the <7GPA group in the presence of >7 GPA group, authors should also mention that ‘from this perspective the presence of a >7 GPA group in the ERP is definitely a positive factor in the performance of <7 GPA group. However, the argument of the authors is not that GPA should not be a determinant factor of performance potential but the point here is that a GPA complemented with intrinsic motivation for research, research self-efficacy beliefs, perceptions of research and curiosity should be considered for selection of candidates for ERP in medical schools’.

If the authors accept the above-mentioned minor revision to the paper, the reviewer recommends that this paper be accepted for publication.

7. PLOS authors have the option to publish the peer review history of their article (what does this mean?). If published, this will include your full peer review and any attached files.

Reviewer #2: No

Reviewer #3: No

---

## [Author Response · Author response to Decision Letter 1]

3 Sep 2021

Dear Pathiyil Ravi Shankar,

We would like to thank you for the opportunity to revise our manuscript. In addition, we would like to thank the reviewers for the helpful review comments, which are greatly appreciated. The comments offer us guidance in the review process and help to improve our manuscript. Below we provide an overview on how we processed the review comments. 

Journal requirements

“Please review your reference list to ensure that it is complete and correct. If you have cited papers that have been retracted, please include the rationale for doing so in the manuscript text, or remove these references and replace them with relevant current references. Any changes to the reference list should be mentioned in the rebuttal letter that accompanies your revised manuscript. If you need to cite a retracted article, indicate the article’s retracted status in the References list and also include a citation and full reference for the retraction notice.” 

Done as requested. 

Reviewer 2

“The comments have been addressed and the project has to be taken forward as per suggestions to be more useful and relevant” 

We would like to thank the reviewer for checking our revised manuscript and are pleased to read that the comments have been addressed. 

Reviewer 3

“This study attempts to throw light on the common oversight of an important benchmark in performance evaluation of medical researchers. The paper is detailed, well explained and has incorporated good statistical tools.” 

We would like to thank the reviewer for reviewing our manuscript and for his/her positive words.

1. Kindly correct grammar inconsistencies at one or two areas in the paper (Eg. Inconsistencies with tenses: Line 273 – ‘All students were included’ is more appropriate as the study has been reported in the past tense throughout the paper)

Done as requested (revised manuscript lines 273, 315, and 318).

“2. If ERP drop-outs were more in the <7 group and this was statistically significant, one can argue that lack of motivation played an important role in the discontinuation from the ERP. Rather than dogmatically defending the findings with two contradictory theories for comparison of two different parameters between the same two study groups (viz. Reflected glory effect and Big fish in small pond theory), the acceptance to the paper can be enhanced with an open-mind approach that is:

- Authors can state in the paper that ‘the higher drop-out rate in <7 group could be justified by the big fish in small pond theory or could be attributed to the lack of motivation in the <7 group. Although the questionnaire shows no difference in motivation between the two groups, the finding of increased drop-outs confounds the result and warrants further research to investigate the reason for increased drop-outs in the <7 GPA group’

- If authors discuss that “Reflected glory effect” explains the increased self-perceived ability of the <7GPA group in the presence of >7 GPA group, authors should also mention that ‘from this perspective the presence of a >7 GPA group in the ERP is definitely a positive factor in the performance of <7 GPA group. However, the argument of the authors is not that GPA should not be a determinant factor of performance potential but the point here is that a GPA complemented with intrinsic motivation for research, research self-efficacy beliefs, perceptions of research and curiosity should be considered for selection of candidates for ERP in medical schools’.”

The reviewer suggests that a lack of motivation played an important role in the discontinuation from the ERP and suggests that we should state in the paper that the higher drop-out rate could be justified by the big-fish-little-pond theory or could be attributed to the lack of motivation in the <7 group. We agree that the big-fish-little-pond theory is applicable here, which is why we mentioned this as a clarification in our discussion section. With regard to motivation, in general it could indeed be the case that motivation decreases and results in drop-out, however, we do not feel that it is justified to draw the conclusion from our data that a lack of motivation in the <7 group may cause the higher drop-out rate. When looking at the levels of intrinsic motivation at both timepoint 1 (table 1) and timepoint three (table 2) we can see that these levels are stable, or even marginally higher at timepoint three, implying that a lack of intrinsic motivation for research does not seem plausible. Especially as our non-inferiority approach shows that the <7 group is not inferior to the >7 group when it comes to intrinsic motivation for research. This view is corroborated by the coordinator of the ERP. We do believe that it could be that apparently motivation does not lead to finishing the programme, which we mention at page 18, line 364-366 of our revised manuscript (“it is remarkable that, though comparable in motivation for research, the drop-out in the <7 group is higher as compared to the ≥7 group, possibly implying that motivation does not lead to ERP completion”). Furthermore, the reviewer suggests that from the perspective of the reflective glory effect the presence of the ≥7 group is a positive factor for the performance of the <7 group, which we agree upon. Indeed, our argument is not that we should exclude the ≥7 group, rather that we should offer the <7 group the chance to participate within the ERP. In line with the reviewer’s suggestions, within the revised manuscript we now mention that ‘from this perspective, the presence of the ≥7 group is a positive factor for the performance of the <7 group’ (page 17, line 358-359).

Again, we would like to express our gratitude to the reviewers for the comments. We are very thankful for the time invested in our manuscript. We really feel it helped us to improve our manuscript considerably, and hope you acknowledge our improvements.

If any questions or concerns remain, please do not hesitate to contact us. Thank you for receiving our revised manuscript, we look forward to our collaboration on this manuscript.

On behalf of all authors,

Kind regards,

Belinda Ommering

---

## [Decision Letter · Decision Letter 2]

11 Oct 2021

PONE-D-21-11278R2The importance of motivation in selecting undergraduate medical students for extracurricular research programmesPLOS ONE

Dear Dr. Ommering,

Thank you for submitting your manuscript to PLOS ONE. After careful consideration, we feel that it has merit but does not fully meet PLOS ONE’s publication criteria as it currently stands. Therefore, we invite you to submit a revised version of the manuscript that addresses the points raised during the review process.

The reviewer has requested some minor revisions to your manuscript. 

We look forward to receiving your revised manuscript.

Kind regards,

Pathiyil Ravi Shankar

Academic Editor

PLOS ONE

Journal Requirements:

Reviewers' comments:

Reviewer's Responses to Questions

**Comments to the Author**

1. If the authors have adequately addressed your comments raised in a previous round of review and you feel that this manuscript is now acceptable for publication, you may indicate that here to bypass the “Comments to the Author” section, enter your conflict of interest statement in the “Confidential to Editor” section, and submit your "Accept" recommendation.

Reviewer #3: All comments have been addressed

Reviewer #4: (No Response)

2. Is the manuscript technically sound, and do the data support the conclusions?

Reviewer #3: Yes

Reviewer #4: Yes

3. Has the statistical analysis been performed appropriately and rigorously? 

Reviewer #3: Yes

Reviewer #4: Yes

4. Have the authors made all data underlying the findings in their manuscript fully available?

Reviewer #3: Yes

Reviewer #4: Yes

5. Is the manuscript presented in an intelligible fashion and written in standard English?

Reviewer #3: Yes

Reviewer #4: Yes

6. Review Comments to the Author

Reviewer #3: Reviewers concerns have been appropriately addressed; the manuscript may be accepted for publication in ite revised format.

Reviewer #4: The present manuscrit is very interesting.

Especially I strongly believe that lower academic curriculum should not be an "exclusion criteria" for further carrer.

Some issues.

Introduction is well written but I think it should be shortened to the focus of this paper, that is the impact of "selection" of students for research

Methods: due to low sample size normal distribution shold be assessed

Methods/results: I think that income of the famly of origin, and the level of education should be appraised as they impact of cv and opportunities

7. PLOS authors have the option to publish the peer review history of their article (what does this mean?). If published, this will include your full peer review and any attached files.

Reviewer #3: No

Reviewer #4: **Yes: **Fabrizio D'Ascenzo

---

## [Author Response · Author response to Decision Letter 2]

3 Nov 2021

Dear Pathiyil Ravi Shankar,

We would like to thank you for the opportunity to revise our manuscript. In addition, we would like to thank the reviewers for the helpful review comments, which are greatly appreciated. The comments offer us guidance in the review process and help to improve our manuscript. Below we provide an overview on how we processed the review comments. 

Journal requirements

“Please review your reference list to ensure that it is complete and correct. If you have cited papers that have been retracted, please include the rationale for doing so in the manuscript text, or remove these references and replace them with relevant current references. Any changes to the reference list should be mentioned in the rebuttal letter that accompanies your revised manuscript. If you need to cite a retracted article, indicate the article’s retracted status in the References list and also include a citation and full reference for the retraction notice.” 

Done as requested. 

Reviewer 3

“Reviewers concerns have been appropriately addressed; the manuscript may be accepted for publication in the revised format”

We would like to thank the reviewer for checking our revised manuscript and are pleased to read that the reviewer suggests to accept the manuscript for publication. 

Reviewer 4

“The present manuscript is very interesting. Especially I strongly believe that lower academic curriculum should not be an "exclusion criteria" for further career.”

We would like to thank the reviewer for reviewing our manuscript and for his positive words. We strongly agree with the reviewer that lower academic performance should not be an exclusion criteria.

“Some issues. Introduction is well written but I think it should be shortened to the focus of this paper, that is the impact of "selection" of students for research”

We agree with the reviewer that the focus of our paper is on selection of students for extracurricular research programmes (ERPs) and we are pleased to read that this came across as such. This study is situated in the context of a bigger problem that needs to be solved, i.e. the physician-scientist shortage, which is one of the reasons many ERPs were implemented. Therefore, we do feel that it is necessary to describe this problem and the reasons for implementing an ERP, logically following into the group that should be targeted to reach the desired outcome (i.e. more students pursuing a physician-scientist career). In this latter phase, questions regarding selection arise for which we investigated students with lower and higher academic performance within the ERP. We do feel, however, that we should keep the bigger picture in mind and provide this bigger picture to the broad readership as well. While still making sure that we did not lose the bigger picture and we did not go against previous reviewer requests to expand some parts within our introduction, we critically went through our manuscript and somewhat shortened the introduction at multiple points. We hope the editor understands our considerations and is satisfied with the current shortening of our introduction. 

“Methods: due to low sample size normal distribution should be assessed”

We indeed assessed normal distribution at the start of our analysis phase. Data was normally distributed. For clarity purposes, we now mention that we established normal distribution of the data explicitly within the ‘analyses’ paragraph (page 11, line 242). 

“Methods/results: I think that income of the family of origin, and the level of education should be appraised as they impact of cv and opportunities”

The reviewer suggests that income and level of education of the family of origin should be taken into account as they might impact cv and opportunities. We agree with the reviewer that these are factors that could certainly play a role in general performance, educational or career choices, and being selected for, for instance, an extracurricular research programme. However, we are afraid we do not have collected these data and are therefore not able to appraise them in the current study. 

Again, we would like to express our gratitude to the reviewers for the comments. We are very thankful for the time invested in our manuscript. We really feel it helped us to improve our manuscript considerably, and hope you acknowledge our improvements.

If any questions or concerns remain, please do not hesitate to contact us. Thank you for receiving our revised manuscript, we look forward to our collaboration on this manuscript.

On behalf of all authors,

Kind regards,

Belinda Ommering

---

## [Decision Letter · Decision Letter 3]

5 Nov 2021

The importance of motivation in selecting undergraduate medical students for extracurricular research programmes

PONE-D-21-11278R3

Dear Dr. Ommering,

We’re pleased to inform you that your manuscript has been judged scientifically suitable for publication and will be formally accepted for publication once it meets all outstanding technical requirements.

Kind regards,

Pathiyil Ravi Shankar

Academic Editor

PLOS ONE

Additional Editor Comments (optional):

Reviewers' comments:

Reviewer's Responses to Questions

**Comments to the Author**

1. If the authors have adequately addressed your comments raised in a previous round of review and you feel that this manuscript is now acceptable for publication, you may indicate that here to bypass the “Comments to the Author” section, enter your conflict of interest statement in the “Confidential to Editor” section, and submit your "Accept" recommendation.

Reviewer #4: All comments have been addressed

2. Is the manuscript technically sound, and do the data support the conclusions?

Reviewer #4: Yes

3. Has the statistical analysis been performed appropriately and rigorously? 

Reviewer #4: Yes

4. Have the authors made all data underlying the findings in their manuscript fully available?

Reviewer #4: Yes

5. Is the manuscript presented in an intelligible fashion and written in standard English?

Reviewer #4: (No Response)

6. Review Comments to the Author

Reviewer #4: (No Response)

7. PLOS authors have the option to publish the peer review history of their article (what does this mean?). If published, this will include your full peer review and any attached files.

Reviewer #4: **Yes: **Fabrizio D'Ascenzo

---

## [Editor Report · Acceptance letter]

10 Nov 2021

PONE-D-21-11278R3 

The importance of motivation in selecting undergraduate medical students for extracurricular research programmes 

Dear Dr. Ommering:

I'm pleased to inform you that your manuscript has been deemed suitable for publication in PLOS ONE. Congratulations! Your manuscript is now with our production department. 

Kind regards, 

on behalf of

Dr. Pathiyil Ravi Shankar 

Academic Editor

PLOS ONE